# Assessment of Global Competence of Engineers for a Sustainable World. Evidence from TA VIE Project

**Isabel Ortiz-Marcos** [1],*, **Luis Ignacio Ballesteros-Sánchez** [1], **Araceli Hernández Bayo** [2], **Rocío Rodríguez-Rivero** [1] **and Gwenaelle Guillerme** [3]

1. Department of Organization, Business Administration and Statistics, Universidad Politécnica de Madrid, C/José Gutiérrez Abascal 2, 28006 Madrid, Spain; luisignacio.ballesteros@upm.es (L.I.B.-S.); rocio.rodriguez@upm.es (R.R.-R.)
2. Department of Automatic, Electrical and Electronic Engineering and Industrial Informatics Universidad Politécnica de Madrid, C/José Gutiérrez Abascal 2, 28006 Madrid, Spain; araceli.hernandez@upm.es
3. Top International Managers in Engineering (T.I.M.E.) Association, 91190 Paris, France; gwenaelle.guillerme@time-association.org
* Correspondence: isabel.ortiz@upm.es

**Abstract:** This paper outlines a contemporary understanding of global competence for engineers, as understood by European engineering companies, and presents the main findings of the Tools for Enhancing and Assessing the Value of International Experience for Engineers (TA VIE) project, launched in 2018. Situational judgment tests (SJTs), or scenario-based approaches were used to measure eleven global competences. Researchers designed the scenarios and contrasted them designing a dictionary of competences containing: the definition of each competence as well as the five levels for each competence (defined by objective behaviours that could be observed). The measurement of competences is performed through a web platform where all data are collected. Nearly 300 students from different countries fulfilled the questionaries, and the results show that students with mobility, have, in general terms, a higher level of global competence than those without international mobility. Communication and flexibility are the competences with higher impact when students enjoy an international experience.

**Keywords:** global competence; internationalization; engineering education

## 1. Introduction

We live in a time of transitions. Our societies are undergoing profound changes, mainly driven by science, in which engineers will have key responsibilities that need new competences [1,2]. Examples include rapid population growth, climate change, industrial expansion, as well as the associated consumption of natural resources and the ever-stronger pressure on the environment in general. This context raised critical questions. What are the competences to meet these global changes? What are the skills to be a global engineer? How does engineering education instil these skills? These questions raise the need to improve and adapt skills. It becomes imperative.

Engineering education has an important role to play. We need engineers not only with technical skills but also with so-called "global competences" [3–5]. We aim at cross-disciplinarily and innovation. People are required to change and innovate "fast", and most innovations require that we are able to link different technologies and competences. We need cross-cultural sensitivity, as people must be able to work in a global environment, both if they work for a foreign company or if they find a job in their home country, seeing that flourishing companies have most of their customers and suppliers abroad. As public higher education institutions (HEIs), we advocate for social responsibility, as we have the mission to improve the world/global quality of life, and we want our engineers to understand the social consequences of their projects [6–8]. For those reasons, it is necessary to strengthen global competences for engineers of the future.

The intercultural or global competences that are more valuable to engineers have been widely studied in the literature [9–11]. However, there is still a lack of studies that allows us to understand the impact of international mobility on engineering student competences with a comparative perspective. This is what TA VIE is meant to assess.

International mobility in higher education, and in this paper, is understood as the experience of studying abroad during one semester, or one or two years. There are several international education programs that help students to have this international experience. The best known is the Erasmus+ program. International mobility allows students to interact with other people from different cultures, helping them to become global citizens.

This paper presents the results of TA VIE project, an Erasmus+ program. The Project help to identify those competences currently essential to work in a multidisciplinary and international context and to measure them. With the joint of those competences (knowledge, skills and attitudes) as a whole, we will obtain the so-called *Global Competence*.

The results of this study are expected to help engineering universities enhance their current engineering curriculum to develop strategies and forms for promoting employability that valorises the competences of engineering graduates with international experience, which will benefit both graduates and their employers, and in the end, the rest of society. Global competence is common to all areas of knowledge and the tool designed may be used in any sector. Nevertheless, the assessment has been performed with engineering students.

We hypothesize that some skills and attitudes improve positively after international mobility. Therefore, to make the best of international mobility, it should be designed accordingly, and its results should be assessed by all those target groups (HEIs, students, employers) involved in the implementation of TA VIE. In consequence, the research questions (RQ) authors want to answer are:

RQ1: Does international mobility strengthen global competence?
RQ2: If yes, how?

## 2. Literature Review

Global and intercultural competence refers to a wide range of skills, knowledge, attitudes, and behaviours. The assessment of some of these components is easier or better established than others [12]. In the late twentieth century, nurturing "global competence" (and related competence) became a goal of education [13–15]. To strengthen global competences, researchers argue that studying abroad is an ideal opportunity [16,17].

The definition and assessment of global competence have been faced by several researchers in different contexts, for example, manufacturing [18] or education [19,20]. In recent years, the European Agenda has increased its focus on the development of new competences required in a global labour market. In 2018, the Organization for Economic Co-operation and Development (OECD) launched the Program for International Student Assessment (PISA) which seeks to investigate the global competences of students in a multi-dimensional way [21]. Researchers have seen with interest this program analysing challenges and the cross-curricular domain presented for the PISA study, suggesting that the challenges faced by it are not sufficiently targeted up to now [22].

Recent research [23] has studied the global competence level of elementary school students in Korea by utilizing the self-reported questionnaire presented in OECD PISA 2018 Global Competence Assessment Frame [21] and has explored the implications of teaching and learning directions related to global competence education. Results suggest the need to design and implement teacher education and training for global competence education based on conducting basic research and comprehensive analysis research.

Other researchers [24] have taken the OECD seriously at their claims around inclusion. They look critically at the global competence framework to ask what PISA means by inclusion and trouble with the idea that inclusion can function effectively within a global standardized assessment. They demonstrate how inclusion takes on new meaning as it moves between each iteration of the global competence framework, showing how

this recontextualization re-orientates inclusion from a social justice imperative toward supporting young people's inclusion into a globalized market economy.

Some authors argue an urgent need to scrutinize the framework that underlies global competence. They present a critical analysis framed by academic literature [25], while others argue how global competence may be a way to renew higher education [26].

Other authors have made an interesting reflection about the domain of intercultural competence in the context of global leadership, showing that it comprised three dimensions—perception management, relationship management, and self-management. Each dimension is characterized by facets that further delineate aspects of intercultural competence [27].

Jung et al. [28] explore the useful contents and methods for global competence development, finding the positive and significant relationship between multicultural acceptability (diversity, relationship, and universality) and global competence.

Kang et al. [29] examine college students' global competence acquisition at a United States and Korean university and assessed the effect of cross- and intercultural online projects implemented simultaneously at both universities. They found that, overall, the projects significantly increased students' intercultural communication skills and knowledge of the other country.

Other authors [30] have shown the importance of integrating mobility in the "protected" context of formal education to go beyond multicultural skills that are necessary for the new professionals needed in Europe.

International mobility has also been presented as an interesting tool to attract international students, looking for the development of internationalised attributes and global competences. It is perceived as an addition to the value of the formal qualification improving employability [31].

The authors of this paper agreed on the definition of global competence by the OECD: "Global competence is the capacity to examine local, global and intercultural issues, to understand and appreciate the perspectives and worldviews of others, to engage in open, appropriate and effective interactions with people from different cultures, and to act for collective well-being and sustainable development" [2].

Teachers and students navigate an increasingly interconnected and interdependent world calling them to create learning experiences to develop global competence [32]. These authors encourage teachers and students to embrace teaching for global competence. In this sense, several initiatives have been carried out in different contexts worldwide [33–36].

The global dimension of engineering has been often incorporated by many academic institutions [1,37,38]. However, at present, researchers highlight the need to integrate environmental and ethical issues to incorporate sustainability competences into the structure of the educational system for future engineers over the long term [39–41]. Engineers participate in international programs and projects and the development of intercultural competence for future engineers is of particular importance, being essential to study the most successful practices in the field [42].

As globalization continues to make cross-cultural interactions more of a reality, the need to develop the cultural competence of students and staff is imperative. The concept of the "global citizen" is particularly pertinent in the higher education landscape [43,44].

Moreover, societies have become multicultural and the need for new tools to communicate effectively across cultures has become fundamental [45]. In a global world, citizens need to open their perspectives and experiences to current world changes.

Education strategies are increasingly aimed at the promotion of sustainability competences [46,47]. Specifically, in the engineering education domain, some previous studies have been conducted to assess global competence [48] and to apply the Miville–Guzman Universality Diversity Scale [49–51]. Student and expert design skills are compared in Atman et al. [52] by applying a playground scenario. In a study by Downey et al. [53], a pre-/post-course study design was carried out based on the proposal of several assessment methods (self-reported items, open-ended scenario question, and multiple-choice knowl-

edge questions). The proposal made by Kilgore et al. [54] implemented scenario-based assessments applied to engineering education.

These previous works have shown that, for several years, there has been a deep concern among HE Institutions to know what usually happens in the international context for engineering students. Competences developed in the international context have been analysed mainly by the EU. Methodologies to measure competences in higher education have been published for academic purposes.

Nevertheless, neither references to the measurement of personal competences (more difficult) nor to tool kits or frameworks have been found.

The International Association of Universities (IAU) launched a survey to understand the process of internationalization to best equip higher education leaders to develop the most effective internationalization strategies (http://www.eaie.org/blog/iau-global-survey/, accessed on 15 September 2021). Nevertheless, the results are based on the quantitative analysis of the results of a survey conducted over the internet.

The tendency is to use quantitative data to measure definite results or outputs, but the method does not seem to be as useful or meaningful at assessing outcomes and the eventual impact. The TA VIE project considers not only indicators but also inputs, processes, and outcomes, collectively.

The TA VIE project has been designed considering all target groups to collect a large variety of points of view and is considered to be innovative since the results want to produce changes that speeds up and improves the way we conceive engineering education internationalization, student training improvement, employability and ultimately, building a better world together.

*Scenarios for Competence Assessment*

Besides identifying the range of skills and attitudes embodied by global competence, an equally important issue deals with how these skills can be best measured.

In the past, self-reports were typically used for assessing people's level on each of the identified skills. However, this self-rating method has inherent shortcomings for competences assessment [55]. One limitation is that self-reports assume subjects possess the necessary self-insight and maturity to rate themselves on each of the global competence components. Another weakness of that approach is that respondents tend to engage in the response which, according to their opinion, will be viewed favourably by others (i.e., effect of socially desirable responding) [56]. Other documented limitations relate to response style bias or reference group bias [57].

These shortcomings in the self-report approach have led to the search for methods that can complement or substitute these types of self-assessment tests. Situational judgment tests (SJTs), or scenario-based approaches, have been proposed as one of the alternatives to the traditional self-report inventories that can overcome the aforementioned limitations [58,59]. SJTs typically present test takers with realistic dilemmas, problems, or scenarios in the context of a situation. These dilemmas are followed by alternative courses of action from which the test taker selects the response based on either the likelihood to perform the action or the effectiveness of the action. Finally, the responses to the SJTs are scored based on a theory-based or an empirical-based approach [60].

These scenario-based methods have a long history of use, dating back to at least the 1920s and have become increasingly popular in personnel selection both in the US and Europe [60,61]. Motowidlo et al. [62] had one of the pioneering contributions of applying SJTs for selecting entry-level managers in the telecommunications industry and showing the validity of this approach to predict the performance of the test takers [59,62].

The research evidence shows that SJTs are a cost-efficient methodology that shows good levels of reliability and predictive and incremental validity over other cognitive and personality tests. As a weakness, most SJTs are context-specific instruments, making it necessary to develop SJTs for specific jobs or contexts [61,63,64].

Because of these previous results, applying the scenario-based method to measure competences has been used during the development of the TA VIE project as a tool to measure global competence.

## 3. The TAVIE Project

"TA VIE project—Tools for Enhancing and Assessing the Values of International Experience for Engineers", co-funded by the Erasmus+ Programme of the European Union, partners collaborate in a strategic partnership of four universities whose focus is on science, engineering and technology (SET): Universidad Politécnica de Madrid (UPM), Ecole Centrale de Nantes (ECN), Budapest University of Technology and Economics (BME), Università degli studi di Trento (UniTrento), and all Members from the T.I.M.E. Association (Top International Managers in Engineering). The chosen countries allow a comparative perspective of different European countries with different educational systems and labour market conditions.

The purpose of the TA VIE project is to assess the importance of global competences and skills required from engineering graduates wishing to work in a multidisciplinary and international context and to measure them. Specifically, the objectives were: (1) identify the global competences (knowledge, skills and attitudes) needed by engineers to work and communicate effectively in organisations and companies characterized by cultural and social diversity; (2) develop a toolkit that helps HEIs, companies and organisations as well as engineers to assess the impact of training and international experience on global competence which will enhance quality assurance of education and international mobility; (3) develop innovative and effective teaching and training strategies for students in higher education, drawing on theory and current best practices, and making better use of the many existing opportunities for embedded mobility and cooperation; (4) develop strategies and ways for promoting employability that valorises the competence of engineering graduates with international experience [9].

This project is deeply innovative since it has been developed as a framework of competences and tools for companies, students and HEI to measure these competences that are strengthened through international experiences. These tools will tune curricula to current and emerging labour market needs. This project connects companies with students with international experience considering and giving value to these strengthened competences.

The ability to measure these competences is an innovation that will increase the knowledge of the impact that internationalization is having on engineering training. This will develop innovative strategies to boost mobility in higher education.

## 4. Materials and Methods

The main goal of the project is to measure the global competences with and without international experience to discover the impact that these mobilities have on engineering students' global competences.

The first step was to identify the most relevant global competences needed. It was performed with companies through 37 semi-structured interviews in five different countries [62]. The framework of global competences for engineers was defined and structured in six clusters and three categories:

- Organizational competences:

A service to the organization (performance-oriented; structured; conscientiousness; service-mindedness; analytical thinking/attention to detail).

- Relational competences:

    ○ B. Cooperation (adaptability; flexibility; collaboration; teamwork; sociability; relation building).
    ○ C. Communication (communication in a foreign language; presentation skills; active listening; humility; empathy; assertiveness; convincing).

    ○    D. Leadership (negotiation; decision making; problem solving; encouraging and motivating others; holistic thinking; long term vision; conflict management; ethical orientation).

-    Individual competences:

    ○    E. Self-knowledge (self-awareness; cultural knowledge; responsibility; resilience).

    ○    F. Proactivity (curiosity; change promotion; oriented to face challenges; acceptance of differences; openness; initiative; creativity).

The competences to be measured (from these 37 previously identified) were selected through the following process: in a first step, a design thinking session with the participation of all partners was developed. During this session, and with the full map of competences, these competences that had appeared in all partners' countries (considering how many companies) were highlighted. The competences with higher ranking were selected. In the second step, it was also considered to include competences from all categories.

At the end of this second step, 15 competences were selected: collaboration; communication; performance orientation; curiosity; analytical thinking; adaptability; problem solving; flexibility; self-awareness; conflict management; oriented to face challenges; resilience; empathy; acceptance of differences and cultural awareness.

SJTs or scenario-based approaches were used to measure these competences. Researchers designed the scenarios and contrasted them with an expert on this methodology. A dictionary of competences was designed containing: definition of each competence as well as the five levels for each competence (defined by objective behaviours that could be observed). Then, scenarios were developed adapting the situation to the international context. All the scenarios have five different reactions proposed to students and they must choose one of them (the one that describes his/her reaction if they have to face the situation proposed). Each choice must be coherent with the levels defined in the dictionary of competences. One example is presented in Appendix A for one competence (acceptance of differences). The competence is defined as well as the five different levels. The scenario with answers linked to levels is shown to clarify.

The measurement of competences is performed through a web platform designed by the TA VIE project, where all data are collected: personal information to characterize the sample and the results of measurement of competences (http://surveytavie.industriales.upm.es/, accessed on 15 September 2021). This first pilot was sent to students to validate the use of the tool to measure these competences. The levels of the scenarios are presented in a random way to guarantee that students do not choose the level considering this. In this first pilot, competences did not appear randomly.

In the validation process of the pilot, 75 students from Spain, Italy and France participated. Answers were analysed in terms of the international mobility experience.

Here, we have a survey, and we are mainly interested in the differences between students with an international experience and those with none. Thus, we have two groups of students, and we are interested in their responses to Qu1 to Qu15. We hypothesize that all the 15 variables are affected by the fact of having or not having international mobility, assuming that better values will be obtained for those students who have international mobility. A multivariate analysis of variance was used to test this hypothesis. Given that the p value is 0.6346, higher than the usual significance levels (0.01, 0.05 and 0.1), we cannot reject the null hypothesis. There is no statistical evidence that the factor (international mobility) affects the multivariate response.

According to Pareto's diagrams, the multivariate response is as follows: selecting six competences that we wanted to check if they together (6-dimensional multivariate response) if they are affected by the difference in international mobility. A multivariate analysis of variance was used to test this hypothesis.

Given that the *p* value is 0.09739 < 0.1, we do reject the null hypothesis for a confidence level equal to 90%. In practice, considering the 6-dimensional multivariate response and

international mobility as a two-level factor; there is statistical evidence that the factor does affect the multivariate response. Each of the six scenarios considered here corresponds to one of the groups we had: A, B, C, D, E and F.

In light of the results of the validation process, some scenarios needed to be reviewed since the statistical analysis showed that they were not measured properly.

In addition, the high number of students that did not finish the survey made researchers consider the need to shorten the scenarios, reducing the number of competences to be measured (the decision was performed considering all the results obtained during the validation process). Was considered the need to present the competences in a random way to avoid the fatigue effect during the reading of the last scenarios.

Four competences were deleted and finally, eleven were kept: communication; performance orientation; curiosity; analytical thinking; problem solving; flexibility; self-awareness; oriented to face challenges; resilience; empathy; acceptance of differences.

Scenarios were reviewed when needed to improve the tool. The design process is presented in Figure 1.

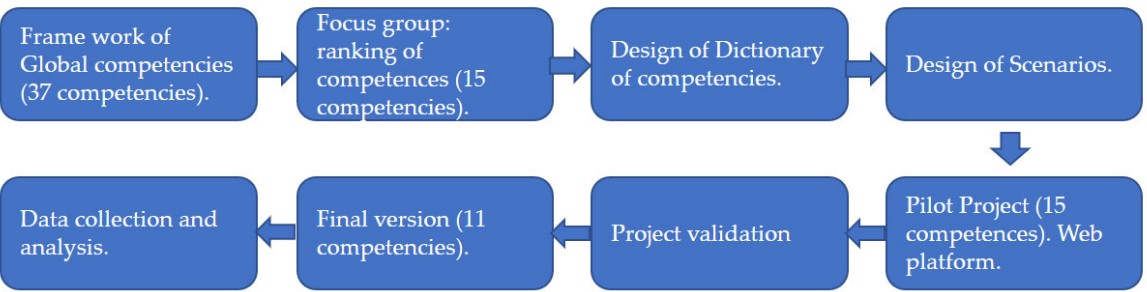

**Figure 1.** Methodology steps.

Analysis of data was performed with descriptive statistics using the software the "R Project". R is a free software environment for statistical computing and graphics.

Histograms are used to represent and compare the results of each level of competence in the different samples (with and without mobility). Mean comparisons using Student's t test for hypothesis contrasts were used for unpaired samples such that the significance of competence between students with and without mobility could be determined.

## 5. Results and Discussion

Data were gathered during 2021, for both students with and without international mobility. This brings the opportunity to compare both groups and obtain relevant results. At this point, it is necessary to mention the pandemic situation due to COVID-19 and the mobility limitations during 2021.

There was a total of 293 valid answers: 77 students were from Spain, 178 from Hungary and 38 from Italy. Concerning gender, 138 were females and 153 males (two selected "other" for their gender); 194 were in a bachelor's degree, 92 in a master's degree and 7 were in a doctoral program (See Table 1).

**Table 1.** Sample features.

|  | Bachelor | Master | Doctoral | Total |
|---|---|---|---|---|
| Female | 90 | 45 | 3 | 138 |
| Male | 102 | 47 | 4 | 153 |
| Other | 2 | 0 | 0 | 2 |
| Total | 194 | 92 | 7 | 293 |

From the total sample (293), 255 had no international mobility and 35 did. The reasons why students have or do not have international mobility is not the subject of this study, but the drop in the number of students undertaking this international experience has been directly affected by the impact of the COVID-19 pandemic and its mobility restrictions, as previous studies have noted [65–67].

Analysis of results by gender and countries was performed, but statistical differences were not found. Analysis considering the impact of international mobility on global competences is shown in Figure 2. Bar diagrams for frequencies considering each score obtained in each global competence for students are presented. "No" and "Yes" answers the question "Did you have an international mobility experience? "No mobility" and "Yes mobility".

A hypothesis contrast for mean comparisons was used to identify possible significant differences between both groups, using Student's t-test (assuming equal variances). Results are presented in Figure 3 and Table 2.

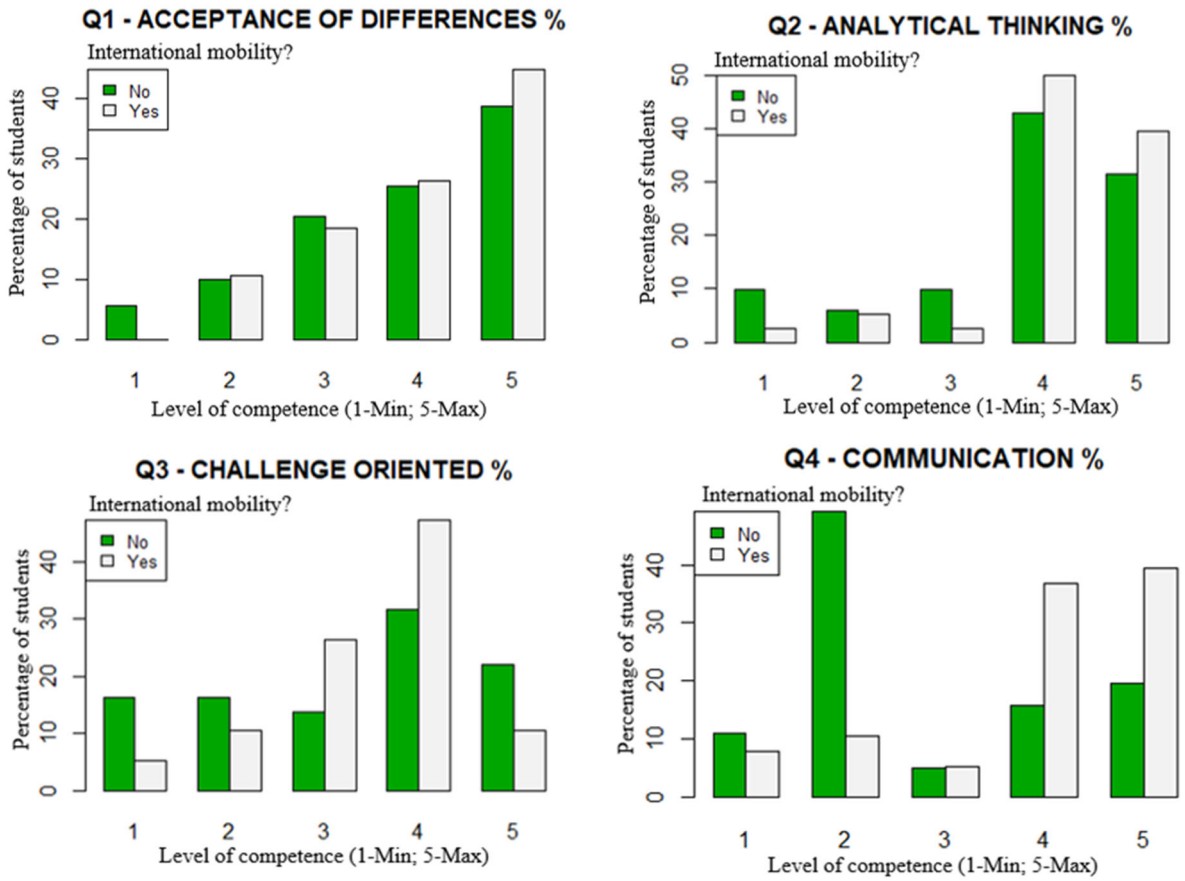

**Figure 2.** *Cont.*

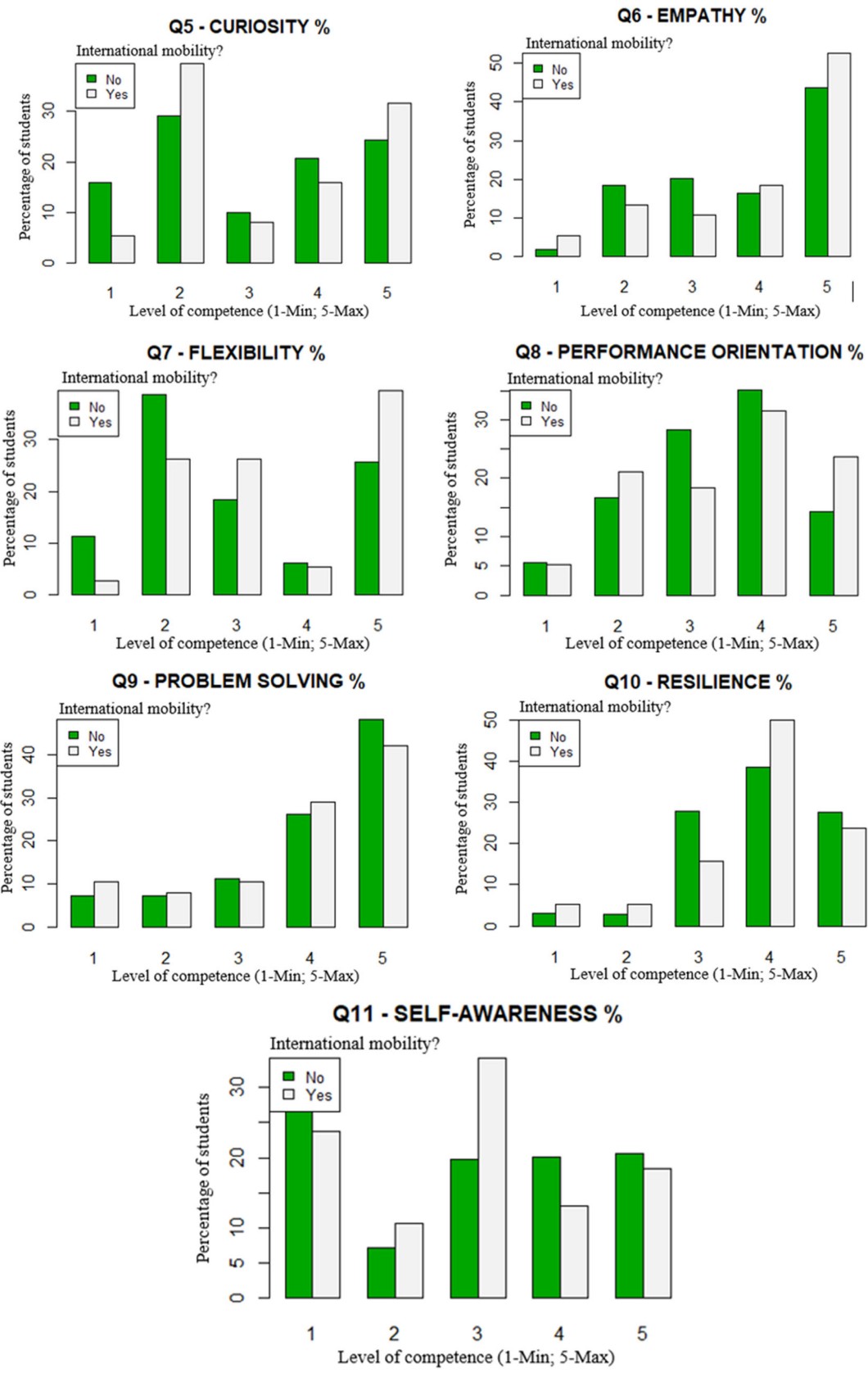

**Figure 2.** Frequency bar charts for global competences measurements.

**Figure 3.** Mean scores for global competences: comparison between mobility and non-mobility group (Yes; No). (* Differences statistically significant).

**Table 2.** Mean scores and comparison using Student's *t*-test.

| Global Competence | Non-Mobility Group | Mobility Group | Dif. | *p*-Value [1] |
|---|---|---|---|---|
| Acceptance of differences | 3.8 | 4.1 | 0.236 | 0.255 |
| Analytical thinking | 3.8 | 4.2 | 0.381 | 0.067 ° |
| Challenge oriented | 3.3 | 3.5 | 0.201 | 0.393 |
| Communication | 2.8 | 3.9 | 1.055 | 0.000 ** |
| Curiosity | 3.1 | 3.3 | 0.206 | 0.415 |
| Empathy | 3.8 | 4.0 | 0.179 | 0.404 |
| Flexibility | 3.0 | 3.5 | 0.566 | 0.019 * |
| Performance orientation | 3.4 | 3.5 | 0.115 | 0.552 |
| Problem solving | 4.0 | 3.8 | −0.170 | 0.436 |
| Resilience | 3.8 | 3.8 | −0.031 | 0.856 |
| Self-awareness | 2.9 | 2.9 | 0.028 | 0.917 |

Note. [1] Student's t-test hypothesis test for paired samples. ° $p < 0.1$. * $p < 0.05$. ** $p < 0.01$.

General results show higher scores for 8 of the 11 global competences for the group with international mobility. For three of them, *analytical thinking*, *communication* and *flexibility*, the differences were statistically significant. For *problem solving* and *resilience*, the difference was negative without significant statistical difference. These results are in line with the argument that studying abroad is an opportunity to strengthen global competences, which has been identified by previous research [16,17]. Thus, international mobility can be a useful learning experience to develop global competence, which has already been identified as a current need [32].

### 5.1. Acceptance of Differences

Acceptance of differences has to do with the ability to acknowledge and appreciate differences between people, cultures and backgrounds.

In this case, students with mobility show higher mean scores than those with no mobility, 3.8 vs. 4.1. No statistical differences were found.

The most frequent answer for both groups was the one with the highest score (5), demonstrating accepting and appreciating diversity, believing that the teamwork outcomes will be more interesting and that learning will be enhanced.

Students, in general, show to be open to different people. In the mobility group, this competence was the one with the second highest scores (4.1) among all global competences. In this same group, no student answered the option with a less score related to avoiding diversity when possible.

Although no clear results were found for this competence and its relationship with international mobility, it is recognized that *acceptance of difference* could be closely related to inclusion approaches [24]. Therefore, more research is recommended in line with the idea that inclusion can function effectively within a global standardized assessment. As well, this competence is related to intercultural competence, which shows to be essential when aiming to develop global leadership and multicultural acceptability [27,28].

### 5.2. Analytical Thinking

Analytical thinking is the ability to solve in-detail problems effectively. It involves a methodical step-by-step approach to thinking that allows to break down complex problems into details and manageable components, taking into account a systemic and global approach.

Students with mobility showed significantly higher scores in this global competence than students without mobility, 4.2 vs. 3.8. This difference was statistically significant at 90% of confidence (*p* value < 0.1).

The most frequent answer for both groups was the one with a score of 4, related to breaking down a complex project step by step into several parts and sharing the findings with team members.

### 5.3. Challenge Oriented

This competence expresses how the person is guided by values and beliefs to achieve important and difficult goals, keeping a sense of focus in difficult situations.

A student with mobility showed minor differences to those without mobility, 3.5 vs. 3.3. No statistical difference was found.

Analysing frequencies of answers, it can be seen that students with mobility are more focused on obtaining the opportunity of having an international experience, which will help them to grow personally and professionally, which is consistent with the motivations of students when travelling abroad.

We would like to underline the importance of this result. When students improve their analytical thinking in an international context, it is expected that they improve their capabilities of integrating sustainability issues, in line with previous needs detected by other authors, specially concerning the education of engineers over the long term [39–41].

### 5.4. Communication

Communication is related to the ability to express one's own thoughts and understanding and interacting with others.

In this case, there are important differences between the scores of both groups. In the group with mobility, students showed higher scores (3.9) than those belonging to the group without mobility (2.8). The difference is statically significant at 99% of confidence (*p* value < 0.01).

Looking at the frequency chart, it can be seen that students without mobility are interested in international mobility and learning from others by listening. However, students

with mobility are more active in the communication process, participating actively, sharing their experience and being curious to hear from other students about their experience.

These results are relevant and in line with previous research [29,44], confirming that having an intercultural experience by studying abroad significantly increases students' communication skills and knowledge of the other countries.

### 5.5. Curiosity

Curiosity expresses how students are interested in new experiences related to something new, willing to dig below the surface, and being interested in many different experiences.

Students with mobility show higher curiosity than those without mobility, 3.3. vs. 3.1. However, no statistical difference was found.

An interesting result is that students without mobility are more goal focused, and students with mobility are more aware of the importance of engineers to be prepared for a global world.

### 5.6. Empathy

This competence demonstrates that a person is aware of other people´s feelings, emotions and social norms, understanding their behaviour and being supportive when needed.

Students with mobility show higher curiosity than those without mobility, 4.0 vs. 3.8. However, no statistical difference was found.

Both groups show, in general, high empathy levels, in terms of being able to make others feel understood.

### 5.7. Flexibility

Flexibility is related to having no trouble changing your own behaviour/opinion.

There is an important difference between both groups. Students from the mobility group showed a higher level of this competence than those within the non-mobility group, 3.5 vs. 3.0. The difference is statistically significant at 95% of confidence ($p$ value < 0.05).

From the answers of the students and the frequency charts, it can be seen that students with mobility accept more and welcome change, seeing it as an opportunity to grow. However, students without mobility, although they are open to change and adapt their behaviour to new circumstances, do not focus on opportunity derived from the change. It is an interesting result obtained since the difference between both groups are clear and aligned with looking at change as something that can help you to grow, rather than something that you have simply to adapt to.

### 5.8. Performance Orientation

This skill represents a strong will to succeed and achieves aims, engaging in multiple tasks and putting in relevant working hours, including strong motivation to use one's own and new resources to achieve goals.

The level of competence demonstrated for both the mobility and non-mobility groups are similar, 3.5. vs. 3.4. No statistically significant differences were found.

Looking at the frequency chart, students with mobility showed more behaviours oriented to high motivation and engagement to actively achieve goals as much as needed, while students without mobility still tended to make efforts to achieve the goal, showing less engagement on performance.

### 5.9. Problem Solving

Problem solving is the ability to analyse a situation, recognize an issue and review the problem features. It includes evaluating how to intervene and find the resources for the most reasonable solution.

In this case, students without international mobility showed higher scores for this competence than those with international mobility, 4.0 vs. 3.8. However, the difference is not statistically significant.

By observing the frequency charts, no important differences were appreciated. Therefore, problem-solving does not appear as a relevant competence when analysing the impact of international mobility on student skills.

Both groups showed high levels of performance in this competence, taking the necessary time and resources to analyse situations, understand problems and intervene to solve them.

### 5.10. Resilience

Resilience is the ability to develop coping mechanisms to withstand and deal with pressure and uncertainty, stress and difficult moments.

No differences in mean scores for this competence were found between both groups, having a global score of 3.8.

Group of students with mobility have more behaviours related to actively looking for resources to face difficult situations and improve their way of performing, and fewer behaviours related to simply listening to others when they experience difficulties in the search for advice and help.

### 5.11. Self-Awareness

This competence is related to being conscious of your feelings, needs and desires, and sensitive to how others may interpret your communication and behaviours.

No differences were found concerning the mean scores of both groups within this competence, with a level of 2.9.

This competence showed to be the lowest among all global competences measured within this study.

Students tend to focus on their external world rather than their internal one for making important decisions. Looking at the frequency chart, students without mobility experience are more focused on job security and economic conditions, while students with mobility are more focused on their career path and future vision of themselves taking into account real opportunities, which is an interesting result.

## 6. Conclusions

This research has allowed us to define and measure relevant global competences of engineering students. The aim has been to analyse whether international mobility has a real impact on students behaviours, demonstrating their skills by using a situational judgment test. Using a final 11 competences test via a web platform, data were gathered and analysed, allowing to compare differences between students with and without international mobility.

Answering research question 1 (RQ1), it is possible to say that students with international mobility have, in general terms, a higher level of global competence than those without international mobility, which is in line with the literature review [16,17], despite the limitations of the sample in a year of restricted mobility.

Regarding research question 2 (RQ2), it can be said that students with international mobility have proven to be more active in the communication process, as well as more flexible when it comes to changing their minds, participating actively and being curious to hear from other students about their experience. These results confirm that having an intercultural experience by studying abroad significantly increases student communication skills and knowledge of other people from different countries, accepting and welcoming change, and seeing it as an opportunity to grow. Besides, an interesting conclusion is that students without mobility are more focused on job security and economic conditions, while students with mobility are more focused on their career path and future vision of themselves taking into account real opportunities.

However, it can be concluded that students can improve their analytical thinking in an international experience, allowing them to take a systemic and global approach. This could have an impact on their capabilities of integrating sustainability issues and confirms partially our research hypothesis: Global competence is strengthened with international mobility, especially for communication, flexibility and analytical thinking competences.

Engineers are key elements in the present and the future of society and the world. Global competences of engineers imply the capacity of building relationships, making the right decisions and contributing to creating a world with equity, justice, inclusion, sustainability and peace. In line with this, international mobility contributes to developing strategic global competences at three levels: individual, relational and organizational.

Conclusions obtained from this study are contributing to a better understanding of the impact and implications of international mobility of engineers. More data should be gathered and analysed shortly such that additional conclusions can be obtained. This will help to define appropriate strategies for the internationalization of universities based on the impact on student competences and learnings.

Some of the main limitations of the study are related to the limited number of respondents to the test (293). As well, there were fewer students in the sample with international mobility (35), partially explained by the impact of COVID-19 on international mobility during 2021. By broadening the sample and obtaining more answers from students from different countries of Europe, it will be possible to include more students with international mobility and obtain possible new relevant insights.

Future research could focus on analysing the main differences of international mobility impact depending on the host university, as well as other factors such as mobility duration. Considering students from other countries out of Europe such as the US, China or Russia or from other fields (not engineering) may give light to improve strategies of internationalization.

In addition, the analysis performed to identify global competences with companies during the development of the TA VIE project could be performed in other areas of knowledge apart from engineering. Comparing results may be of interest.

**Author Contributions:** Project leader, I.O.-M.; team participating in UPM, L.I.B.-S., A.H.B. and R.R.-R.; from TIME Association, G.G. helped with tool diffusion. All authors have read and agreed to the published version of the manuscript.

**Funding:** European funding: ERASMUS+ Action 2, project number 2018-1-ES01-KA203-050477.

**Institutional Review Board Statement:** Not applicable.

**Informed Consent Statement:** Not applicable.

**Data Availability Statement:** Not applicable.

**Acknowledgments:** The authors would like to acknowledge the European Commission for funding the project that made it possible to work together on this research. We acknowledge the partners, all the companies, and the HEIs that participated by contributing their knowledge and time.

**Conflicts of Interest:** The authors declare no conflict of interest.

## Appendix A. Example of Scenario for ACCEPTANCE OF DIFFERENCES Competence

Competence Definition: Acknowledges and appreciates differences between people, cultures and backgrounds.
Levels:
Level 1: He/she avoids diversity when possible.
Level 2: He/she is performance focused without considering diversity as a barrier.
Level 3: He/she accepts some but not all diversity, i.e., different backgrounds.
Level 4: He/she trusts on others to accept differences.
Level 5: He/she accepts and appreciates diversity, believing that the teamwork outcomes will be more interesting and that learning will be enhanced.

Scenario: You have been selected to study in China during the coming academic year, where you are enrolled in a negotiation course. There are students from different countries and backgrounds (not just engineers) enrolled in this course. The professor asks students to divide themselves into teams to work together:

(a) I normally prefer to work with other students from my own country. I know them and I know it will be easier to work together and we will achieve better results. (Level 1).

(b) I will choose the best students to achieve the best results in terms of negotiation. (Level 2).

(c) I am willing to work with students from other countries, but I prefer students with a similar profile to mine irrespective of other issues. I know that it´s easier for people with a similar outlook to understand each other and build something together. (Level 3).

(d) I trust that working with students from other countries, will benefit myself and the team performance. Results will be better even if we need more time to understand each other. (Level 4).

(e) I am open to working with others, and I look for students from other countries, of different gender and from different backgrounds to form the group. (Level 5).

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
