# Peer review of "Assessment of Global Competence of Engineers for a Sustainable World. Evidence from TA VIE Project"

_sustainability, doi:10.3390/su132212924_

Round 1
Reviewer 1 Report
Every year, hundreds of thousands of people in Europe take the opportunity to study abroad or work on European projects supported by the EU (Europe Union). In this context, comparing ideas and good practices based on solid fact-finding and first-hand experience has been a central component of European cooperation in education. This approach has developed in many ways, from academic networks, study visits and partnerships of numerous kinds to the EU's policy-making role today in such central issues as defining quality indicators and the future objectives of education and training systems.
The paper presents the results of the "TA VIE Project - Tools for Valuing and Assessing the Values of International Experience for Engineers", co-funded by the European Union Erasmus+ Programme, launched in 2018. The authors define and measure the relevant global competencies of engineering students and identify, using a study of quantitative nature, the most relevant ones comparing students who have done mobility with students who have not done mobility.
Strengths:
The paper presents an excellent organization, and the points advanced in logical ways.
This paper traces the intellectual progression of the research field and cites Related work.
The documentation of sources and references are appropriate.
The tables and figures are helpful and appropriate.
Weaknesses
The authors could have talked about the innovation of the study, the limitations/applications of the research and the practical and social application.
It would be important to understand if the identified global competences are common to all areas of knowledge or if they are specific for engineering students. I think they are common in all areas. However, the sample used was engineering course students.
The framework may also be subject to further validation in other contexts.
Author Response
Thank you for your comments. Paper has been reviewed considering them. Answer to all of them is sent in attachment.

Reviewer 2 Report
Dear author/s
Τhank you for giving me the opportunity to read your manuscript.
I read your manuscript with great interest.
However, the manuscript needs significant/major or minor changes that can be significantly improved. Below you will find some major or minor points in the manuscript which needs clarification, refinement, reanalysis, rewrites or/and additional information and suggestions for what could be done to improve it.
From the section 1 (Introduction) the aim or/and objectives of the study or/and hypotheses or/and research questions are absent or/and unclear, and which should be numbered and clearly written. To help you, I quote some questions (as list of points) so that it can be included in your introduction:
-What is the importance of making this research/contribution that it brings to the literature in the field?
-Why should readers be interested?
-What problem/ gap resolve/fill this research?
-To fill this gap (resolve this problem) what solution/intervention/benefits does this research bring? (in other words, how the proposed study will remedy this deficiency/gap/problem and provide a unique contribution to the literature).
-What is the research question which address to the purpose of the research?
From the section 4 (Materials and Methods) are missing some points and information, e.g., the type of methodology, information about the methods you used, the pilot phase, the reliability, ethics issues, the consent protocol, measuring instrument, the sample selection criteria, etc.
Although some of them are mentioned, a better reorganization / presentation may need to be done, and of course with a relevant reference in the literature where needed.
Most importantly, the type of methodology you follow and the type of analysis should be mentioned. You need to be clear and unambiguous.
Moreover, it would be good if you could mention further demographics of your research sample.
Also, the program/software you used to analyze the data is not mentioned.
In summary, this section needs some minor revision for a better presentation.
Section 5 (Results and Discussion) presents only the results, which are complete and with good presentation. The results are not discussed in this section.
Based on the name of your section, you should discuss your results (i.e., compare the results of other researches, correlate the results with information from and the through the literature review, etc.).
The discussion of the results can be included in a new section as a "Discussion" and this section can be named "Results".
In summary, please expand this section (i.e., Results and Discussion), and compare your results to the ones found in similar studies. In particular, please cite more of the journal papers published by MDPI where possible.
Section 6 (Conclusions) is very simple and look poor. Please expand this section.
Kindly read your manuscript again with a clear mind and make the necessary corrections. Also, kindly check for grammatical errors, and new publications that could form part of the manuscript.
As a final comment, I recommend an in-depth revision of your manuscript.
Author Response
Thank you for your comments, they help us to improve our work. Paper has been reviewed considering them. The answer to all of them is sent in attachment.

Reviewer 3 Report
The article investigates how the international mobility of students can improve their holistic engineering skills. As a member of an institution where study abroad is compulsory for all students, and I can observe its effects, I am happy to see a more systematic investigation.
However, the paper needs to clarify several points. First, the concept of international mobility is not defined. Under what circumstances did the students answer yes to the international mobility question. Any foreign travel, or strictly just university exchange programs? When a student has no mobility, what does that mean? The lack of financial resources, or simply having no desire for work/study abroad? Since this is the main criterion for grouping the observations, it is essential to give more information.
Mentioning the COVID-19 situation raises more questions. How did the pandemic affect the project? As the influence on the observed data can be decisive, detailed considerations are required.
Description of the scenarios for competence assessment, or at least some examples of them, are sorely needed. It is left to the reader's imagination to come up with these scenarios. What is a score of 1, and what is a score of 5?
The captions for the figure should contain enough information that the reader can understand the bar charts without consulting the text. It is slightly disruptive for the reading that the bar charts come before the descriptions of the competencies.
The statistical methods are not included. The same problem holds for the used statistical software package.
The paper seems to distinguish between competence and competency but does not distinguish between the two meanings.
Line 347 has a copy-paste error, hinting at the lack of thorough checking before submission.
Author Response

(The authors gave the same response as above.)

Reviewer 4 Report
Dear Authors,
Please find below my review report.
During my initial documentation for this review, I tried to found similar previous works in the literature.
I found the following reference: https://doi.org/10.3390/su12229568 and it seems there are many paragrahps in your manuscript that are very similar to the previous one.
The similar sequences from your manuscript are at rows 14-16, 50-53, 60-63, 68-72, 104-108, 114-132, 364-366.
Please revise this aspect and clarify these similarities: if the article is a "remake", then explain it. If not, I suggest you to revise the above mentioned paragraphs.
The proposd title of your article is too general, compared to the content. In your manuscript you discuss about a very specific educational project "TAVIE project– Tools for Enhancing and Assessing the Values of International Experience for Engineers".
I recommend you to adapt the title in order to mirror the content of the article: a suggestion would be "Assessment of Global Competence of Engineers for a sustainable world - Evidence from TAVIE project". Or something like this.
In last part of the Abstract, you say that "Authors encourage engineers to strengthen these global competences through mobility experience to contribute to building a more sustainable world."
I recommend you to revise this sentence because it is a personal opinion of the authors. A scientific article should make recommendations based only on the research results.
I didn't find the research question of your manuscript. The research question should appear in the Introduction section, based on research gap from the previous literature.
The Literature Review section should be improved in order to clarify the general context of your research. Here I recommend you to cite the following valuable resources: https://doi.org/10.24818/ie2020.02.01, https://doi.org/10.3390/admsci11040105 - learning tools and higher education (rows 75-80 in your article), https://doi.org/10.3390/admsci10040096, https://doi.org/10.3390/admsci11030103 (rows 124-132 in your article).
Between section "2. Literature Review" and section "3. The TAVIE project" there is no visible relationship. Please include a specific sequence where you describe the "link" between the literature review and the TAVIE project.
At rows 219-222 you have: "Finally, 15 competencies were selected: Collaboration; Communication; Performance orientation; curiosity; Analytical thinking; Adaptability; Problem solving; Flexibility; Self awareness; Conflict management; Oriented to face challenges; Resilience; Empathy; Acceptance of differences and cultural awareness."
Then, at rows 246-249 you have: "Four competencies were deleted and finally, eleven were kept: Communication; Performance orientation; curiosity; Analytical thinking; Problem solving; Flexibility; Self awareness; Oriented to face challenges; Resilience; Empathy and Acceptance of differences."
Please describe the criteria used to delete the mentioned 4 competencies, because at this moment it is not very clear how you reduced the number of competencies.
Before the "Materials and Methods" section, I recommend you to define your research hypotheses, so that the readers know what you want to test and to discover in your research.
Row 309: you have a typo-mistake: "p-vale <0.1". It should be "p-value <0.1".
Before the Conclusions section, I recommend you to insert a Discussion section. Here you should present your results and compare them with some previous results from the literature.
In the Conclusions sections you should include your main findings, the limitations of your research (limited area of respondents, limited number of respondents etc.) and the future research directions.
Good luck!
Author Response

(The authors gave the same response as above.)

Reviewer 5 Report
Thank you for the opportunity to revise this interesting paper. I think the research addresses an important topic. However, there are aspects that the authors could reconsider:
The introduction does a good job at presenting the topic and research questions, but it does not explain all the paper does. For example, it does not describe what the TA VIE project is, leaving the reader with an incomplete picture of what the paper is going to show.
The discussion of the findings could also be developed more. The authors present their quantitative results, but this is not linked at all to previous studies or theories. It would be important to connect the current study to previous one to show how this study fits within the established literature, if it confirms or not previous findings, or extends the field or the theoretical framework is builds on and in which way.
Finally, please discuss the limitations of the study, and especially which areas of future research arise from your study that researchers interested in the topic could address in the future.
Author Response

(The authors gave the same response as above.)

Round 2
Reviewer 2 Report
Dear author/s
I have read again with much interest your revised manuscript.
The manuscript has been significantly improved.
Please change / correct
line 510 "Answering RQ1 we can [...]", and
line 512 "Regarding RQ2 we conclude [...]"
Kindly check for grammatical errors, and new publications that could form part of the manuscript.
Author Response
Thank you for your comments. They have been considered and help us to improve.

Reviewer 3 Report
Thank you very much for implementing the changes.
just two minor issues: the competence vs. competency made a comeback in plural: competences vs competencies
Line 299 does not make sense.
Author Response
Thank you for your comments. They have been considered (explained in attachment). They help us to improve.

Reviewer 4 Report
Dear Authors,
I have read the new version of your manuscript and I can say it is improved, compared to the previous one.
However, it still needs improvements because the Sustainability Journal is a high-ranked one and the scientific requirements are very specific.
From my point of view, the article has two major weaknesses: the literature review section and the research hypothesis issue.
These two aspects should be solved according to the below recommendations.
As I already told you in my previous remarks, the Literature Review section should be improved in order to clarify the general context of your research.
Here I recommend you to cite the following valuable resources: https://doi.org/10.24818/ie2020.02.01, https://doi.org/10.3390/admsci11040105 - learning tools and higher education, https://doi.org/10.3390/admsci10040096, https://doi.org/10.15240/tul/001/2021-03-006, https://doi.org/10.3390/admsci11030103.
Regarding the research hypothesis, you say at rows 70-71: "Our hypothesis is that some skills and attitudes improve in a positive way after an international mobility".
But, at rows 292-293 you say: "In this case, the answers are 15 dependent variables, and our hypothesis is that all together are affected by the difference in international mobility"
These two hypotheses seem to be different.
Please clarify this aspect.
Also, I didn't clearly find the confirmation (or not) of the initial research hypothesis.
You should allocate a special section for this issue in the Result & Discussion chapter of your manuscript: please provide an applied discussion and a relationship between your results and the initial research hypothesis.
Kind Regards!
Author Response

(The authors gave the same response as above.)

Reviewer 5 Report
I have no further comments for the authors
Author Response
Thank you for your positive feedback.
Round 3
Reviewer 4 Report
Dear Authors
I appreciate the changes that you made in the article and I consider that you managed to make a better version of it, fulfilling the basic criteria necessary for a scientific paper.
Kind regards
Author Response
Thank you very much for your comments. They helped us to improve the result of our work.
